# Population Genetics for Inferring Introduction Sources of the Oriental Fruit Fly, *Bactrocera dorsalis*: A Test for Quarantine Use in Korea

**DOI:** 10.3390/insects12100851

**Published:** 2021-09-22

**Authors:** Hyojoong Kim, Sohee Kim, Sangjin Kim, Yerim Lee, Heung-Sik Lee, Seong-Jin Lee, Deuk-Soo Choi, Jaeyong Jeon, Jong-Ho Lee

**Affiliations:** 1Animal Systematics Laboratory, Department of Biology, Kunsan National University, Gunsan 54150, Jeonbuk, Korea; veritas72@kunsan.ac.kr (S.K.); ksj050505@naver.com (S.K.); 2Animal & Plant Quarantine Agency, Gimcheon 39660, Gyeongbuk, Korea; mycomania21@korea.kr (S.-J.L.); dschoi@korea.kr (D.-S.C.); dollmock@korea.kr (J.J.); acarologist@korea.kr (J.-H.L.)

**Keywords:** exotic pests, haplotype network, invasion route, microsatellite, Southeast Asia, tracing origin

## Abstract

**Simple Summary:**

The more the global agricultural product trade becomes active every year, the more foreign pests’ invasion probability increases. Accordingly, many notorious invasive pests are spreading worldwide, and the nations should try to block their introduction through quarantine systems. As an important quarantine pest, the oriental fruit fly, *Bactrocera dorsalis* (Hendel) (Diptera: Tephritidae), is one of the most destructive pest insects for fruit crops in tropical and subtropical areas. This species is a highly invasive and economically important pest with a broad host range. Here, we collected 40 geographically or temporally different collections from 12 Asian countries, including four from the Korean border quarantine detection, and performed haplotype analysis and population genetics analysis.

**Abstract:**

To infer the introduction sources of the oriental fruit fly, *Bactrocera dorsalis*, we used a mitochondrial marker to reconstruct the haplotype network and 15 microsatellite loci to reveal genetic structure and relationships between the geographically or temporally different collections from Asia. We performed Approximate Bayesian computations to infer a global origin and a source of the quarantine collections found in Korea. As a result, the 40 populations were divided into three groups, of which genetic similarity is not related to the geographic vicinity. Korean samples had a similar genetic structure to Taiwan and Thailand ones. Our results suggest that the place of origin of the *B. dorsalis* specimens found in Korea’s border quarantine is likely to be Taiwan or Thailand. As the global origin of *B. dorsalis*, we estimated that Taiwan and Thailand were most likely the global origins of Southeast Asian populations by testing hypothetical scenarios by the approximate Bayesian computation analyses. Our results will allow easier identification of the source region of the forthcoming invasion of quarantined *B. dorsalis* specimens.

## 1. Introduction

In the 21st century, both transportation and communication developed rapidly around the world. The development of various means of exchange has facilitated the movement of human and material resources across national borders [1]. However, due to multiple forms of international exchanges, such as trade and overseas tourism transnational marriage, invasive pests from foreign countries are also increasing [2,3,4]. Moreover, climate change as a global problem has promoted invasive pests’ settlement, and the threat and economic loss of indigenous ecosystems by alien creatures are increasing [5,6]. Moreover, it is expected that most of the Palearctic temperate regions, including the Korean Peninsula, will be changed to a subtropical climate by the end of the 21st century [7,8]. Among the world’s invasive pests, the oriental fruit fly is considered the most critical insect that faces these problems by introducing exotic species [9].

The oriental fruit fly, *Bactrocera dorsalis* (Hendel), is classified into the family Tephritidae of Diptera. Surprisingly, this species is taxonomically a very complicated taxon since *B. dorsalis* complex *sensu lato* contains 75 described species [10]. These fruit flies are considered economically important pests because they have strong invasiveness and a wide range of hosts, mostly fruits [11]. Although *Bactrocera dorsalis* had been a taxonomically indeterminate group, *B. dorsalis*, *B. papayae sensu stricto* (*s. str.*) and *B. philiepinensis* were recently classified as the same biological species by Boykin et al. [12]. Correspondingly, Schutze et al. [13] also considered that *B. papayae*, *B. invadens* and *B. dorsalis*
*s. str.* were synonyms of *B. dorsalis*. In this study, we focused on *B. dorsalis* redefined by the previous studies [12,13] for tracing the origin of the invasive source.

The oriental fruit fly is ecologically adapted to tropical and subtropical climates [14]. A mated *B. dorsalis* female punctures mature fruit’s skin with its ovipositor and it deposits her eggs [15]. Fruit-fly eggs hatch into larvae after 2–4 days in summer or 10–20 days in winter [16]. The larvae hatched from the eggs feed on fruits, which are damaged by decay or falling out, thus losing the fruit’s commercial quality. The third instar larvae exit the fallen fruit and burrow into the soil under the host plant to pupate [17]. After one or two weeks, the adult insects that came from the pupa begin mating [18].

Some previous studies predicted that the oriental fruit fly cannot overwinter in the Korean Peninsula’s harsh winter temperatures because it generally lives in tropical climates [14,19,20]. However, the Korean Peninsula’s temperature is rising year by year [21,22,23], which is the same phenomenon that occurs throughout East Asian countries [24,25,26]. This problematic situation may soon allow fruit flies to invade and settle, causing damage to the fruit industry. As in other temperate regions such as Europe, it is no longer convinced that East Asia and Northeast Asia’s temperate areas are safe from the invasion of oriental fruit flies [19,20,27,28,29]. With regard to the ancestry origin of *B. dorsalis*, some previous studies suggest that it was first originated from the Mainland China or Taiwan [30,31,32,33,34] and may have migrated to the west and spread to Southeast Asia [30,35,36], while the other studies showed that India, Nepal and its neighboring countries are the most ancestry origin of *B. dorsalis* spreading to the South and East Asia [10,37]. These conflicting theories regarding the ancestry origin of *B. dorsalis* have not yet been clearly concluded.

The oriental fruit fly is known as a quarantine pest that is very important to be alert to the quarantine in the world. Likewise, in the Republic of Korea, *Bactrocera dorsalis* (Hendel) is currently included in the prohibited pest list of the plant protection law, and thus Animal and Plant Quarantine Agency (APQA), Ministry of Agriculture (MoA) [38], is strengthening quarantine activities to prevent the introduction of the fruit fly. This species is considered a very dangerous pest because it has a short life cycle, a high reproductive potential, a broad host range of over 250 fruits and vegetables [10,39], and is expected to cause significant economic damage when invading new regions [37,40]. The oriental fruit fly has invaded and damaged Japan, China and Taiwan [41,42,43,44,45], of which countries are geographically close to the Korean Peninsula. Especially in Japan, 43 million USD was spent eradicating the oriental fruit fly for 18 years [43,45]. If the oriental fruit fly invades Korea, the damage caused by export restrictions is estimated to reach up to 200 million USD estimated by MoA [46]. It is most important to quarantine the oriental fruit fly to prevent invasion. Therefore, if the species is detected in quarantine or investigation process, it is essentially needed to shut off from the invasion pathway by proceeding with an expeditious trace of the origin.

To trace the origin of introduced pests, the microsatellite DNA markers are a useful tool for population genetic and molecular ecological studies. The sequences of microsatellites are short tandem repeats that are widely distributed throughout the entire eukaryotic genome [47]. According to Mendelian genetics, the microsatellite loci selected for population genetic analysis are transmitted to offspring, neutral and polymorphic [48]. Microsatellite markers generally have short tandem repeats with fast mutation rates and present a co-dominant feature, unlike mitochondrial markers or other nuclear DNA markers [49]. Therefore, microsatellite data can be a useful tool for determining intra- and interspecific relationships [50]. Several studies were designed to study the population genetics of highly invasive tephritid fruit flies (e.g., *Ceratitis capitata* [51,52,53], *Bactrocera dorsalis* [54,55,56,57], *Zeugodacus cucurbitae* [58,59] and *Bactrocera oleae* [60,61,62]). These developed microsatellite markers can be used for analysis in other closely related species [49,63]. In related species, microsatellite DNA sequences’ flanking regions tend to be conserved. Therefore, microsatellites cross-amplification of related species is often used as an alternative approach. In this study, the useful microsatellite markers were selected previously developed from the closely related species of *B. dorsalis*, which can display the sufficient resolution of the within-population level.

This study was aimed to trace the origin and invasion route of the *B. dorsalis* individuals detected in the quarantine process. Samples of the *B. dorsalis* were collected from the countries within native or invasive ranges, having active trade and a similar climate to the Korean Peninsula, and were used for population genetics analysis. Fifteen useful microsatellite markers were tested and selected from previous studies [64,65,66,67,68] to analyze the genetic relationships and construct the allelic database that can identify quarantine or invasive individuals’ origin. In detail, there are three main objectives: (1) identify the genetic structure of *B. dorsalis* in East Asia countries; (2) conduct genetic comparisons between the samples obtained from South Korea and East Asian countries; and (3) infer the East Asian origin and invasion source of *B. dorsalis* collected by quarantine activities in South Korea.

## 2. Materials and Methods

### 2.1. Taxon, Sample Collection and DNA Extraction

In this study, sampled collections all have not been obtained in prohibited areas such as national parks where permission is requested, thus we make it clear that there is no content in relation to permission of collecting samples. We obtained 565 *B. dorsalis* individuals from 36 geographically or temporally different (if in the same location) collections and four quarantine collections, which included the regionally collected populations in 12 East Asian countries between 2015 and 2017 (Figure 1 and Appendix A). Among them, 565 individual samples comprised 90 individuals of six collections in Taiwan, 49 of three in China, 49 of three in Vietnam, 72 of three in Laos, 25 of one in Myanmar, 45 of four in Thailand, 174 of 12 in Philippines, 24 of one in Malaysia, in addition to single samples from Cambodia, India and Nepal, respectively (Figure 1 and Appendix A). In addition, 26 samples assigned to the quarantine-detected Groups 1, 2, 3 and 4 were collected from South Korea (Figure 1 and Appendix A). The individuals of 26 Korean quarantine-detected samples were assigned to four groups (KR 001-004) by STRUCTURE analysis (*K* = 4) in Appendix A. Although these quarantine-detected groups as pooled non-natural ones were not like other naturally collected populations, we set them as four ‘virtual’ populations to infer their origins and invasion routes compared with foreign samples according to the STRUCTURE analysis (Appendix A). To capture *B. dorsalis*, we equipped a transparent cylindrical acrylic trap with methyl eugenol and insecticide-infused cotton, and we installed this apparatus near the host plant.

We identified the samples of *Bactrocera dorsalis* as the same biological species, *B. dorsalis* complex, consisting of *Bactrocera dorsalis* s. str., *B. papayae*, *B. philippinensis* and *B. invadens* [12,13]. Accordingly, ‘*Bactrocera dorsalis*’, which is mentioned after that, is meant to the valid species taxonomically revised by the two recent studies [12,13]. After we collected the samples, we identified them as belonging to the species complex and used them in the experiment.

First, we morphologically identified the collected fruit fly samples based on the morphological characteristics, especially on veins of wing and body patterns [69,70]. Second, we experimented with molecular identification, using the barcode region of the cytochrome *c* oxidase subunit I (*COI*) [71] the samples we identified as *B. dorsalis*. After processing the morphological and molecular identifications, we performed the following haplotype and population analyses, using the exact identified *B. dorsalis* samples.

To be used in the DNA experiment, we cut one or two hind legs from each fly individual, considering the specimen’s size and condition. We performed the DNA extraction according to the manufacturer’s protocol of LaboPass™ Tissue Genomic DNA mini Kit (COSMOGENETECH, Seoul, Korea) or AccuPrep^®^ Genomic DNA Extraction Kit (BIONEER, Daejeon, Korea).

### 2.2. Haplotype Network Analysis

We performed a polymerase chain reaction (PCR), using the extracted DNA template. The site we amplified was *COI* barcode, of which the length of the amplification product was 658 bp [71]. The primer set of LepF1 (5′-ATTCAACCAATCATAAAGATATTGG-3′) and LepR1 (5′-TAAACTTCTGGATGTCCAAAAAATCA-3′) was used to amplify the *COI* barcode sequence [71]. We used AccuPower^®^ PCR PreMix K-2037 (BIONEER, Daejeon, Korea) for the PCR, we mixed 2 µL of extracted DNA with 1 µL of 10 pmol forward primer, 1 µL reverse primer and 18 µL nuclease-free water to amplification. We performed PCR, using a GS482 thermo-cycler (GENE TECHNOLOGIES, Essex, UK) according to the following procedure: initial denaturation at 95 °C for five min, followed by 38 cycles of 95 °C for 20 s; annealing at 45 °C for 30 s; extension at 72 °C for 40 s, and a final extension at 72 °C for 5 min.

We visualized the PCR products with electrophoresis on a 1.8% agarose gel with low range DNA ladder to check for positive amplifications. For electrophoresis, we prepared a mix of 1 g SeaKem^®^ LE Agarose (LONZA, Bend, OR, USA), 0.8 g Metaphor™ Agarose (LONZA, USA), 3 μL of RedSafe (LONZA, Bend, OR, USA) and 100 mL of 1×TAE buffer (BIONEER, Daejeon, Korea) gel. We ran 3 μL of PCR product and 3 μL of 100 bp DNA ladder (BIONEER, Daejeon, Korea) for 25 min at 100 V, using Mupid^®^-ex (TAKARA BIO, Kusatsu, Japan). We used a gel image analysis system WGD-20 (DAIHAN, Seoul, Korea) to visually confirm the amplification products. We sequenced successfully amplified samples, using Applied Biosystems (ABI, Waltham, MA, USA) 3730XL DNA Analyzer by MACROGEN, Inc. (Seoul, Korea).

We used the CHROMAS 2.4.4 (TECHNELYSIUM Pty Ltd., South Brisbane, QLD, Australia) and MEGA X [72] for sequence analysis and alignment. We saved the aligned sequence as a FASTA format file that we used in the MEGA X program. We performed haplotype analysis, using the 465 *COI* sequences by DNASP 6.0 [73]. We reconstructed median-joining (MJ) networks between the *COI* haplotypes, using NETWORK 5.0.1.1 [74] to infer the haplotypes evolutionary relationships.

### 2.3. Population Genetics Analysis

#### 2.3.1. Microsatellite Marker Screening and Design of Multiplex PCR Set

We used random DNA samples to investigate the amplification success and polymorphism of microsatellite markers and retrieved 43 candidate microsatellite loci (MSL) with 86 primers from the five previous studies [64,65,66,67,68] to find and apply MSL microsatellite loci (MSL) with high efficiency and resolution. In the preliminary experiment, we used 43 candidate MSL to four random populations for testing amplification. We conducted a total of 172 PCR tests, and then we estimated the amplification success and polymorphism of each marker. Finally, we selected 15 MSL (Appendix A), which we subsequently screened for primer-dimer formation with Multiple Primer Analyzer (THERMO FISHER, Seoul, Korea). Based on the screening results, we designed four multiplex PCR sets that were made by a combination of fluorescent dyes, such as FAM, VIC, NED and PET (ABI, Waltham, MA, USA), at the 5 ‘-end of the forward primer (Appendix A).

#### 2.3.2. Multiplex PCR and Fragment Analysis

We performed multiplex PCR for each primer set, using the extracted DNA of *B. dorsalis* sample. For multiplex PCR, we used AccuPower^®^ PCR PreMix K-2037 (BIONEER, Daejeon, Korea). We mixed the 2 µL of template DNA and 2 µL of set mixture solution and 18 µL of nuclease-free water, and we performed the PCR at a total volume of 20 µL. We performed the PCR, using a GS482 thermo-cycler (GENE TECHNOLOGIES, Essex, UK) according to the following procedure: initial denaturation at 95 °C for 5 min, followed by 40 cycles of 95 °C for 20 s; annealing at 53 °C for 30 s; extension at 72 °C for 40 s, and a final extension at 72 °C for 5 min. We conducted a fragment analysis-standard run (500LIZ size standard), MACROGEN, Inc. (Seoul, Korea), using fluorescent-labeled (5′ modification) forward primers, which we successfully amplified by checking the gel image.

We performed allele reading, using GENEMAPPER ver. 4.0 (ABI, Waltham, MA, USA). We converted raw allelic datasets to a text form, and then it was analyzed in GENALEX 6.503 [75] working in Microsoft Excel 2019 (Microsoft, Redmond, WA, USA).

#### 2.3.3. Data Analysis

Using 565 *B. dorsalis* individual samples from 40 collections (Figure 1 and Appendix A), we performed the data processing for calculating allele frequency and genetic distance, principal coordinates analysis (PCoA), heterozygosity, *F-*statistics, polymorphism, allelic patterns and Nei’s genetic distance, using the GENALEX 6.503 [75,76]. We used the GENCLONE 2.0 [77] to identify multilocus genotypes (MLGs) among populations and to calculate the genotypic diversity (*GD* = [*G*/*N*]), where *G* is the number of different MLGs and *N* is the sample size [78]. Observed and expected heterozygosity (*H_o_* and *H_e_*) values among loci were estimated by using GENEPOP 4.6 [79,80,81]. Using the Bonferroni correction sequential for all trials with multiple comparisons, Hardy–Weinberg equilibrium [82] and linkage disequilibrium [83] tests were performed on the adjusted levels of significance. The deviations from HWE was estimated for heterozygote excess or deficit. We used FSTAT 2.9.3.2 [84] to estimate the gene diversity (*H_s_*), the mean number of alleles (*N_a_*), the number of effective alleles (*N_e_*) and allelic richness (*R_s_*). We also calculated the pairwise genetic differentiation (*F_st_*) values [85,86].

We used BOTTLENECK version 1.2.02 [87] to detect the effect of a recent bottleneck in the populations in our samples. We used the strict stepwise mutation model (SMM) and the two-phase model (TPM) considered as appropriate for microsatellite datasets. A model includes both 90% SMM and 10% TPM for 10,000 iterations. Significant deviations in observed heterozygosity among all loci were tested by using a nonparametric Wilcoxon signed rank test [88]. Due to small sample sizes, the bottleneck analysis excluded the several groups with a population of fewer than four samples, which were KR 004 (2 individuals), CN SHA (1), KH PNP (1) and NP POK (1). Therefore, we excluded five individuals in four groups, and we normally conducted the analysis.

We used STRUCTURE 2.3.4 to analyze the genetic structure of 565 *B. dorsalis* individuals from 40 populations, using a Bayesian methodology [89]. We set the number of assignments (*K*) from one to 15 and conducted five replicate runs for each *K* value. In each run, it was performed by Markov chain Monte Carlo (MCMC) of 500,000 repetitions with an admixture model after a burn-in period of 30,000 steps. We obtained the value of delta *K* (Δ*K*), using the ad hoc quantity, on the basis of the rate of the second order of the likelihood change [90]. We calculated *K* by using the online resource STRUCTURE HARVESTER [91] that explained the data structure to perform this process correctly. We conducted STRUCTURE results visualization, using DISTRUCT 1.1 [92]. GENECLASS 2 was used to carry out the Bayesian tests with optioned to detect first-generation migrants [93]. For each population, the program estimated its likelihood of belonging to any other population or to the place where it was collected. The most likely source of the sample has the highest assignment probability on the assigned genotype. Bayesian method was used to calculate allelic frequencies of population [94] with Monte Carlo resampling for estimating the significance of assignments (simulated individuals = 10,000, alpha = 0.01).

We tested independent groupings based on biogeography with analysis of molecular variance (AMOVA) by ARLEQUIN 3.5.1.2 [95,96], with its significant value estimated with the nonparametric permutation method designed by precedent research [97]. We used POPULATIONS 1.2.30 to estimate the relationship between groups, using the analyzed allele values, and created a population-distances phylogenetic tree of intergroup genetic structures and explore migration areas [98]. For this analysis, we used a file in the format and the alleles coded with three digits of GENEPOP.

Approximate Bayesian computation (ABC) was carried out in DIYABC 1.0.4 [99] to estimate the relative likelihood of alternative introduction scenarios of the oriental fruit fly, using microsatellite datasets. The DIYABC test includes genetic admixture events in introduced populations, serial or independent introductions and the comparison of complex scenarios involving bottlenecks [100]. For modeling scenarios, the parameters consist of the rate of admixture, the duration of the bottleneck during colonization, the effective number of founders in introduced populations, the stable effective population size and the times of split or admixture event [101]. To estimate the posterior distribution of parameters and to select the most likely scenario, the program produces a simulation dataset [101]. A simulation dataset generated by the program is used to choose the most similar to the observed (selected) dataset (*n_δ_*) and then used to calculate the posterior distribution of parameters [99]. We analyzed to test two different ABC analyses, using all or partial microsatellite data. We performed the DIYABC analysis to track the ancestral populations and the migration process among all experimental groups. Eventually, conducted the DIYABC 1.0.4 analysis to determine which of the groups we collected was the most ancestral group and which group was derived. Three virtual groups were first set up to proceed with the analysis. We referred to the STRUCTURE (*K* = 3) results and PCoA to set the virtual group (see Results). Above all, we removed the Korean quarantine samples from all groups, and then we regrouped the remaining samples into G1 (Subgroup 1), G2 (Subgroup 2) and G3 (Subgroup 3). The G1 consists of the samples of China, Vietnam, Laos and Malaysia; the G2 consists of those of Taiwan and Thailand; and the G3 consists of those of the Philippines, India, Myanmar, Ho Chi Minh and Vietnam. In the first analysis, we analyzed six scenarios with G1 G2 and G3 origins to find the most ancestral group and the topology between them (Appendix A). In the second analysis, we analyzed nine scenarios for tracing the Korean quarantine groups (Appendix A). A million simulation datasets were obtained for each scenario. A generalized stepwise model (GSM) as the mutational model for microsatellites was used, which assumes increases or decreases by single repeat units [99]. To identify the posterior probability of these three scenarios, the 1% simulation datasets are selected for the logistic regression, 0.01% ones for the direct approach as closest to the pseudo observed dataset [101]. From the simulated and observed dataset, the summary of statistics was calculated for each of the tested scenarios, such as genetic differentiation between pairwise groups (*F_ST_*), classification index, the mean number of alleles per locus (*A*), mean genetic diversity within- and between-group, Goldstein distance and shared alleles distance (*D_as_*).

## 3. Results

### 3.1. Haplotype Network

A total of 465 *B. dorsalis* sequences on *COI* barcode were generated in this study (GenBank accession nos. MW322099–MW322563). We built a crown-like MJ network, and in total, 50 haplotypes (H1–H50) were obtained (Figure 2). There were some high-frequency haplotypes (such as H1, H2, H3, H23, H32 and H31) located in the center, and other rare haplotypes were connected to them through several mutation steps. In most cases, it was connected radially from the highest frequency haplotype, H1, through several mutation steps. Several haplotypes were found as connected by mutations converging from two different haplotypes. For example, H5 was a converged mutation haplotype of H2 and H23, while H2 and H23 were haplotypes of divergence mutations that occur in different directions from the dominant haplotype, H1. Besides, H11 was a mutation haplotype converging from H3 and H25, and those were both connected with H1. H14 (CN) and H38 (PH) were further isolated with eight steps from H1.

The H1 as the highest frequency haplotype included 312 individuals comprised of the samples found in all countries, which means H1 commonly exists in all countries (Appendix A). As the secondarily highest frequency haplotypes, both the H31 and H32 consisted of individuals only from the Philippine population. The Korean quarantine individuals, the samples included in pop KR 001-4, were found in haplotype H1, H2, H3 and H4; in particular, H4 consists of two Korean quarantine individuals. We used a total of 34 individuals from Taiwan in the analysis, and 11 haplotypes were shown. We used 117 individuals from the Philippines in the analysis, and we observed 19 different haplotypes. The Philippines showed a wide variety of haplotypes, but the haplotype diversity of Taiwan seemed to be higher than that of the Philippines in relation to the population size (Appendix A).

### 3.2. Genetic Differentiation within and between Populations

In this study, we genotyped 565 *B. dorsalis* samples by using fifteen microsatellite loci, which all were discovered to be non-clonal MLGs, i.e., non-identical genotypes estimated by multiple loci (Table 1). Therefore, in all groups, the MLGs were the same as the number of individuals in each population. Besides, each individual had a MLG that was not shared with the other, so every *GD* value was 1.00. The observed heterozygosity (*H_o_*) and heterozygosity (*H_e_*) values from all collected East Asian populations ranged from 0.378 to 0.704 and from 0.200 to 0.751, respectively (Table 1 and Appendix A). In HWE, there was no population deviation due to heterozygote excess or deficit (*p* < 0.0001). Gene diversity (*H_s_*), the mean number of alleles (*N_a_*) and the number of effective alleles (*N_e_*) averaged 0.70, 6.30 and 3.70, respectively, and allelic richness (*R_s_*, mean ± SD) was 2.69 ± 0.23.

We estimated pairwise genetic differentiation (*F*_ST_) between 37 different geographical populations (Appendix A), excluding CN SHA, KH PNP and NP POK. Pairwise comparisons of *F*_ST_ values showed that four Korean quarantine groups (KR 001-004) were genetically close to Taiwan populations (TW JIA, KAO, TIC, TIN and TIP) with mean *F*_ST_ as 0.028, to the Laos population (LA VI1-3) with the mean *F*_ST_ as 0.098, and subsequently to Thailand populations (TH BAN, CHA, DA1 and DA2) with the mean *F*_ST_ as 0.020. Taiwan and Thailand’s populations were geographically distant, but they were genetically close to each other. The mean value of *F*_ST_ between Taiwan and Thailand’s populations is 0.045. However, the Korean and Philippine populations’ genetic distance was the largest, with the mean *F*_ST_ value as 0.135. The mean *F*_ST_ values of each group were 0.082 in KR, 0.013 in TW, 0.020 in CN 0.046 in VN, 0.004 in LA, 0.000 in MM, 0.020 in TH, 0.000 in KH, 0.000 in IN, 0.000 in NP, 0.036 in PH and 0.000 in MY.

As the results of BOTTLENECK [87], a significant observed heterozygosity excess (*p* < 0.05) was not detected from the Wilcoxon sign-rank tests (SMM and TPM), whereas the shifted mode was observed in the seven groups: KR 001, KR 003, TW JIA, TH BAN, TH CHA, IN DEL and PH BUS. This seems to show a bottleneck process occurred in the population, even although the bottleneck test should be cautiously regarded because the sample size for some populations was less than 30 individuals [88].

### 3.3. Genetic Structure and Assignment

We show the results of the PCoA in Figure 3. The *x*-axis is the value of coordinate 1 (−1.20 to 0.60), and the *y*-axis value is in the range of coordinate 2 (−0.60 to 1.20). Groups with similar values of the coordinates 1 and 2 are located close to the graph plane. Twelve countries are represented in the legend (Figure 3). The 39 groups were grouped into three masses we labeled as G1, G2 and G3. The KR, which means individuals found by quarantine in Korea, was included only in G1 and G2. The G1 comprises groups of CN, VN, LA and MY, and KR 003. The G2 consists of TW and TH and included two Korean quarantine groups, KR 001 and 002. Members of the G3 belonged to the first group of VN, IN, MM and PH groups but did not include the Korean quarantine group. The quarantine group KR 004 did not belong to any group, and KH and NP also did not belong to any group because they had very low values of coordinate 1.

In all STRUCTURE analyses, the most likely number of clusters was *K* = 2; using the Δ*K* calculation of the Evanno et al. [90] method result (Appendix A), we found that the best value was 317.32 on Δ*K* = 2, the second-best value 186.73 on Δ*K* = 3 and the third-best value 31.24 on Δ*K* = 6 (from *K* = 1 to *K* = 15). Although *K* = 2 was the best, the results of *K* = 3 and *K* = 6 were referred thereafter because those were effective to display the relationships between the populations when compared and combined with the other analyses such as PCoA (Figure 3). The STRUCTURE analysis for all samples resulted in two distinct clusters, the structure of the group was divided into LA similar types and TW similar types (Appendix A). The Laos type population (green component dominance) includes two Korean quarantine groups (KR 003, 004), China (FJS, SHA and YUN), Vietnam (HCM), Laos (VI1,2,3), Myanmar (YAN), Cambodia (PNP), India (DEL), Nepal (POK), Philippines (GUG, GUZ, LU1-2, MIN and PA1-5) and Malaysia (PEN). On the other hand, Taiwan’s similar type (with the dominant red component) includes two Korean quarantine groups, KR (001,002), Taiwan (JIA, KAO, TIC, TIN and TIP), Vietnam (BC1,2), Thailand (BAN, CHA and DA1,2) and the Philippines (LU3). Some groups (VT HCM, MM YAN, KH PNP, IN DEL, NP POK, PH BUS, GUG, GUZ, LU1-2, MIN and PA1-5) that belonged to the Laos type [102] at *K* = 2 were separated into a third group at *K* = 3 (Appendix A). The STRUCTURE analysis for all samples resulted in six distinct clusters, and the Philippine population showed a very mixed pattern of genotypes (Appendix A, and Figure 3). The Korean quarantine group KR (001,002) had a genetic pattern similar to that of Taiwan. In contrast, quarantine group KR (003,004) had similar genetic patterns with Laos; LA (VN1,2,3) and China; CN (FJS) and Malaysia; MY(PEN) (Appendix A, and Figure 3).

The assignment test using GENECLASS 2 (Table 2) showed the average probability with which samples were destined to the most likely reference population. The self-assignment probability values (SAPV) averaged 0.418 ± 0.273 (mean ± SD) in all populations, and in the Korean quarantine group, SAPV averaged 0.468 ± 0.05. In the Taiwanese population, SAPV averaged 0.260 ± 0.112, whereas it averaged 0.480 ± 0.451 in China, 0.288 ± 0.142 in the Vietnamese population, 0.203 ± 0.048 in Laos, 0.522 ± 0.293 in the population from Thailand and 0.390 ± 0.184 for the populations in the Philippines.

Five cases to confirm genetic variance between the preordained groups were analyzed by using AMOVA implemented in ARLEQUIN [95]. There is no significant apportionment of variance between the Korean vs. East Asian country, within populations in Case 1 (*p* = 0.2176). Genetic variance of about 4.57% among groups in Case 2 (Based on the STRUCTURE (*K* = 2)) and 6.13% among groups in Case 3 (Based on the PCoA and STRUCTURE (*K* = 3)) suggests that there are relatively different regional structures between the compared regions. In Cases, 1, 4 and 5, there was no significant genetic variation between groups (Table 2). The analysis of the Case 1 clearly shows that the KR quarantine group is not an indigenous species or an invasive species that have settled for a long time, showing 0.61% of among group percentage of variation. Through the Case 4, it was confirmed that the genetic variation according to the collection period was more significant than the regional genetic variation, but there was no seasonal effect with a value of 3.28%. Analysis of the Case 5 (based on the longitude 113° E, west side vs. east side) showed 1.56% of the variation. Because of this, it was not possible to confirm the genetic variation for regional isolation.

### 3.4. Inferring an Introduction to Test Hypothetical Scenarios by ABC Analysis

Most of the forty populations were regrouped into three subgroups by referring to the results we obtained with the PCoA and structure (Figure 2 and Figure 3 and Appendix A). Each of the three regrouped subgroups formed a more extensive group containing populations with similar genetic structure. Subgroup 1 has a Chinese allelic pattern and is composed of individuals from China and Vietnam (Cuc Phuong) as ‘CH + VN’. Subgroup 2 has a Taiwan-like allelic pattern and consists of Taiwan and Thailand objects as ‘TW + TH’. Subgroup 3 is a set of allelic patterns similar to the Philippines, composed of groups from the Philippines, Myanmar and India as ‘PH + MY + IN’.

Analyses ‘A’ constructed and analyzed six candidate scenarios to infer the ancestor relationship of ‘CH + VN’, ‘TW + TH’ and ‘PH + MY + IN’ (Appendix A). In Scenarios 1 and 2, ‘CH + VN’ was designed as an ancestor group (Appendix A); in Scenarios 3 and 4, ‘TW + TH’ was designed as an ancestor group (Appendix A); and in Scenarios 5 and 6, ‘PH + MY + IN’ was designed as an ancestor group (Appendix A). We observed the highest logistic regression value in Scenario 4, and the next highest value we observed was in Scenario 3 (Table 3 and Appendix A). As a result, both Scenarios 3 and 4 implied that ‘TW + TH’ was an ancestor group (Appendix A).

We performed the analyses ‘B’ of DIYABC to trace *B. dorsalis*’ invasion origin into the Korean Peninsula. For this reason, we performed this analysis by adding KR 001, KR 002 and KR 003 as the fourth subgroup into the cladistics relationships of analyses (A), in which ‘TW + TH’ set to the most basal group due to the results of their ancestry (Table 4 and Appendix A). In the closest logistic regression, Scenario 3 had the highest values in Analyses B-1 and B-2 (Table 4 and Appendix A). This means that the two Korean groups, KR 001 and KR 002, were originated from ‘TW + TH’. In Analysis B-3, Scenario 1 showed the highest value of posterior probability. As a result of this analysis, KR 003 was originated from ‘CH + VN’.

## 4. Discussion

### 4.1. The Genetic Structure and the Global Origin of Bactrocera dorsalis

*Bactrocera dorsalis* is inferred to be a species native to Southeast Asia, and it is spreading worldwide and occurring in many countries due to its strong invasiveness [11]. The main distribution areas in Asia are Bangladesh, Bhutan, Cambodia, South China, Hong Kong, India, Indonesia, Japan Ryukyu Islands, Laos, Malaysia, Myanmar, Nepal, Ogasawara Islands, Pakistan, Philippines, Sri Lanka, Taiwan, Thailand and Vietnam [14]. However, the species is not confined only to Asia, and it was recently reported to have been introduced to North America, specifically in Hawaii, California and Florida [103]. Especially in the United States, it is now found in all major Hawaiian Islands after a sudden invasion in 1944/1945 [104]. The case of fruit flies invading California shows that it is challenging to exterminate this species because invasion and eradication were frequently repeated between 1960 and 2007 [11,105].

Moreover, the oriental fruit fly also has a history of invasion and settlement in the Mariana and Tahiti Islands in the Pacific and the African Continent [106,107]. Several studies have argued that the *B. dorsalis*’ genetic structure and isolation-by-distance tend to be weakly supported within Asia [28,54,57,108], even though several patterns are evident in our study.

Our PCoA and STRUCTURE analyses show that the fifteen microsatellite markers we used in this study were able to separate a total of 565 samples from 40 populations into three main Subgroups (Figure 3 and Figure 4). There was a genetic variation between the three subgroups: Subgroup 1 was defined as the Chinese type, Subgroup 2 as the Taiwanese type and Subgroup 3 as the Philippines type (Figure 3). According to other previous research papers, invasion processes from tropical Asia to non-Asia have been observed, so finding the Asian invasion origin can help researchers find the global spread source of *B. dorsalis*. Between nations, border division cannot be an isolation factor for each species’ populations. For example, Vietnam’s territory has a longitudinally elongated shape along the coastline. Because of this territory’s shape, populations VN BC1 and VN BC2 collected in Northern Vietnam had genetic structures similar to Yunnan in China than VN HOC, belonging to the same country (Figure 4). The VN HOC population from Southern Vietnam was more similar in genetic structure to KH PNP, a group in Phnom Penh in Cambodia, which is geographically adjacent. The case of the group IN DEL in Delhi, India, and the sample NP POK in Pokhara, Nepal, are also inferred from this phenomenon. On the other hand, the Laos populations, LA VI1, LA VI2 and LA VI3, were collected with a unique strategy. The three groups from Laos were regularly collected in the same place, using the methyl eugenol trap [109]. During the sampling period from 2016 to 2017, there was no introduction from other groups in the area. Therefore, we can infer that the three Laos populations do not compose a metapopulation but continuously maintained populations presumed to be isolated (Table 2 and Figure 4) [110].

As mentioned in the results, between some groups, the genetic structure was similar according to the natural habitat range rather than the administrative division of humans (Figure 3 and Appendix A) [47]. However, other groups deviated from this expectation. India, Nepal, Myanmar, Cambodia and Southern Vietnam had similar genetic structures to the Philippines, even though they were not located in adjacent regions (Figure 4). The Laos populations’ genetic structure was identical to those of Malaysia, Fujian and Western China. Thailand’s *B. dorsalis* populations’ genetic structure is similar to those from Taiwan that are as much as 2400 km away across the sea. As this result suggests, one can assume that the migration, i.e., a transference of individuals affected by human activities, between groups with similar genetic structures occurred. Wu et al. [30] made the following arguments through mitochondrial *NDI* gene sequence analysis. Generally, ancestors show significantly more genetic diversity than derivative populations because of the founder effect [111]. According to this principle, it was considered that the Chinese *B. dorsalis*’ populations originated from Southeast Asia (Manila, Pattaya and Bangkok) [30]. Symmetric migration patterns were detected in Yunnan, Guangdong, Fujian and Taiwan, while the asymmetric migration of gene flow indicated multiple invasions and multiple origins. The detection of gene flow, using mitochondrial genes, strongly supports our results [30].

As the previous studies on the spread of oriental fruit flies argue, the first cause of *B. dorsalis* migration is the food transportation by humans and tourists’ movement along the countries’ borders [30,37]. It may be secondarily caused by typhoons’ generation and its activity in the northern hemisphere occurring every year [112,113]. As the tropical cyclone between 180° E and 100° E in the northern hemisphere, typhoons often transport tropical pests to subtropical or temperate regions along their path [112]. This area is called the Northwestern Pacific Basin and is the most active tropical cyclone basin on Earth, where one-third of the annual typhoons are generated here [114]. Generally, typhoons created in the Northwestern Pacific Basin move northwest [115]. However, when they reach 20° to 30° north latitude, they change their routes northward or northeastward due to the westerlies’ effect and crosses Northeast Asia [116]. Many studies revealed that insects living in tropical regions were transferred to subtropical or temperate regions because of typhoons. Otuka et al. [117] suggested a non-intentional *B. dorsalis* transmission by typhoons from the Philippines to Okinawa, Japan. Shoji [118] recognized that typhoons were an essential factor in moving butterflies from the Philippines to the Ryukyu Islands. Typhoons are suggested as a cause of unintentional introductions of other pests, as well. In our study, it was impossible to trace the exact reason for the international spread and migration of *B. dorsalis*. However, the weakly revealed *B. dorsalis*’ genetic flow direction is similar to frequent typhoons’ routes, such as from Taiwan to Jeju island.

There were several hypotheses on the global origin of *B. dorsalis*. Three sources of the Chinese *B. dorsalis* specimens were proposed by classical studies [35]. Since *B. dorsalis* was first recorded in Taiwan [31], Wang [31] hypothesized that this species was introduced from Taiwan about a century ago to Mainland China and spread to other Asian and Pacific countries in the next 90 years. Nevertheless, Wang [31] hypothesis was not supported by the microsatellite diversity data [32,33]. Interestingly, a previous study [33] that used nine microsatellite markers suggested that Guangdong, China or Southeast Asia (Cambodia, Laos and Thailand) were the mainland fruit flies’ source based upon their two-way migration estimates, which is similar to our results. The study of Shi, Kerdelhue and Ye [34], based on the high levels of local genetic variability, proposed Yunnan as a possible source area for the *B. dorsalis*. If not, they argued that at least the Yunnan area would be of old colonization [119]. Chen, Ye and Mu [36] estimated that *B. dorsalis* on the island of Taiwan and Hainan might have migrated from Mainland China, crossing the Taiwan Strait and the Qiongzhou Strait. This migration was also due to the increase in the amount of trade in agricultural products such as fruits and vegetables in recent decades [35]. These previous studies suggest that Mainland China or Taiwan may be the origin of *B. dorsalis*. Some of them may have migrated to the west and spread to Southeast Asia [35,36]. Similarly, the previous result of Wu et al. [30] suggested that oriental fruit flies migrated from Eastern Asia, from places such as Taiwan and China, and spread to the western regions, such as Thailand or Laos. Accordingly, our study estimated that *B. dorsalis* spreads and migrates from the Southwestern Asia, near Thailand, to China, Taiwan and the Philippines, and from there, it repeatedly reinvades or migrates to Northeast Asia, from Taiwan and the Philippines (Table 2 and Figure 4). However, based on the scenario of *B. dorsalis* migration from west to east by Clarke et al. [10] and Qin et al. [37], it was suggested that India, Nepal and their neighboring countries are, most likely, the ancestry origin of *B. dorsalis*. In this study, there is a weak side to the verification of these hypotheses because sampling from India and Nepal was relatively insufficient. Therefore, further study is needed to clarify the debate of its origin to expand our genetic analysis with the addition of samples from India and Nepal.

### 4.2. Inferring Source Population for Korean Quarantine Samples

Although many similar studies have been performed before, it was challenging to trace the introduction source of *B. dorsalis* Asian population [33,120]. Researchers could deduce that Northern Southeast Asia or Southern China were likely to be the introduction source because the gene flow from Southeastern China or islands to inland was detected [28,30,33,35,54,108]. Wu et al. [30] attempted to reveal the genetic structure and origin of *B. dorsalis* by using the mitochondrial *ND1* gene. Network analysis performed by mitochondria identified the migrations between China, Taiwan, Thailand, Laos and the Philippines. As a result, Wu et al. [30] detected the migration between Thailand and Taiwan. These results support why the microsatellite structures from Taiwan and Thailand specimens, despite being geographically far apart, are similar in our study.

Recently reported studies have suggested that the theory of origin in China and Taiwan, which has been supported, results from a lack of sampling in previous studies [121]. These studies suggest India and Bangladesh as new invasion origins [37,121]. However, as a result of our analysis, populations from India were subordinated to those from the Philippine genetic group, and both the Taiwanese and the Chinese populations constituted a separate pool of alleles (Appendix A and Figure 3). As a result of the DIYABC analysis, the Taiwanese group was confirmed and located in the ancestral position (Table 3 and Appendix A).

This study’s most crucial aim was to answer the question about the invasion source of some *B. dorsalis* obtained at Korea’s quarantine border. Among the samples we obtained, KR 001 and KR 002 had similar genetic structures to Taiwan or Thailand (Figure 3 and Figure 4). On the other hand, KR 003 had identical populations and genetic structure in Fujian, China or Laos (Figure 3 and Figure 4). Although K004 was also similarly clustered to Fujian, China or Laos (Figure 4), it could not be estimated from where it originated due to large difference genetically from other groups (Figure 3). As a result of this, it is estimated that *B. dorsalis*’ origin, which can be introduced through the Korean Peninsula’s borders, may be various countries (multiple cases), not a single country. The genetic structure found at the highest frequency in the border quarantine sample was from Taiwan or Thailand, and 19 individuals belonged to it (Figure 3 and Appendix A). Following that, seven samples were of Chinese or Laos type (Figure 3 and Appendix A). Considering the fruit trade scale and annual tourist entry rate regardless of the geographical accessibility, Thailand and China is more likely to be the introduction source than Taiwan and Laos. Unfortunately, even if compressed into these four countries, it is difficult to pinpoint a specific country for the source of introduction in this study.

### 4.3. Applications for Future Quarantine

The tropical and subtropical fruit industries generate income and employment in many countries, became the primary means of foreign currency income and provided a balanced diet internationally regarding human health and nutrition [82]. Therefore, for developing countries in tropical and subtropical regions, the fruit industry must be a project that cannot be given up for the national economy’s vitality [82,122]. It is a vibrant sector with progressive expansion in production, international trade and consumption [82]. With the development of food processing technology and trade, the tropical fruit industry shows an increase of 3.5% annually, and the significant fruits with significant growth rates include bananas, mangos, pineapples and papaya [123,124,125]. These fruits are the primary *B. dorsalis*’ host plants [11], and Asia is the largest tropical fruit production region, with annual production accounting for 66% of global production [125].

In the 1990s, the Korean fruit trade was opened, and tropical-fruit imports have increased significantly since the 2000s, due to the Free Trade Agreement’s conclusion. Most tropical fruit imports depend on Asian countries, and their imports increase each year (data from Korea International Trade Association). In addition, the number of foreign visitors or foreign residents increased from major Asian countries due to tourism, study abroad, employment, international marriage, etc. (Statistics Agency data). For this reason, the Korean government spurs the APQA to prevent the invasion of foreign pests and eradicate invaded ones.

The invasion risk of *B. dorsalis* is increasing through the channels of portable food, portable belongings and trade stuff. The temperature is rising each year worldwide and in the Korean Peninsula as well, changing from a temperate climate to a subtropical climate. With this phenomenon, if tropical pests invade the Korean Peninsula, it will be fatal to the fruit industry. The geographic range in which an insect or plant lives is naturally dynamic and varies over time. Human-induced climate change, civil engineering and trade are rapidly destroying and rebuilding the insects’ and plants’ habitat barriers [126,127,128]. According to its high score in the invasion risk assessment estimation and its serious hazard upon the fruit industry, the APQA, Korea, has defined *B. dorsalis* as a ‘prohibited pest’. In this respect, our study will be applicable to elucidate *B. dorsalis*’ source of introduction, providing support for future quarantine and management actions regarding *B. dorsalis*’ invasions by both the Korean government and the Korean fruit industry.

## Figures and Tables

**Figure 1 insects-12-00851-f001:**
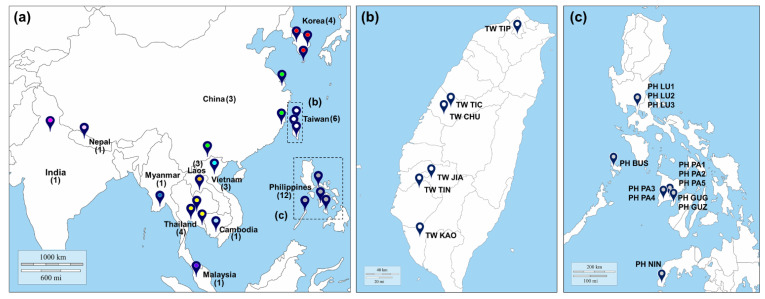
Sample collection sites of *Bactrocera dorsalis* in Southeast Asia. Map of Southeast Asia showing the *B. dorsalis* sample sites collected for this study. (**a**) East Asia map for sampling. Number of collections from each country indicated in parenthesis. (**b**) Collection area on Taiwan Island. (**c**) Collection areas on the Philippine Islands. Detailed location information is described in Appendix A.

**Figure 2 insects-12-00851-f002:**
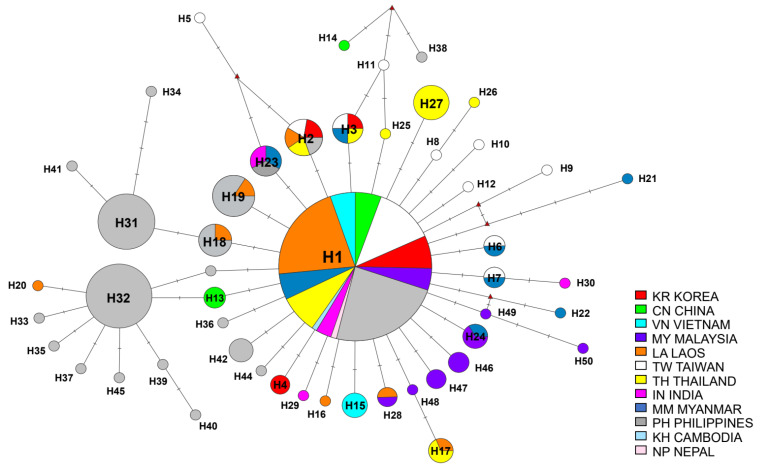
Median-joining haplotype network of 465 *B. dorsalis* sequences based on *COI*. The size of nodes and pie segments were proportional to haplotype frequency. The H1, containing 312 sequences, includes the biggest number of sequences. Each segment color indicates the country sample sources included in the pie. The branch’s length separated by cross marks on the nodes are proportional to the number of mutational changes between haplotypes.

**Figure 3 insects-12-00851-f003:**
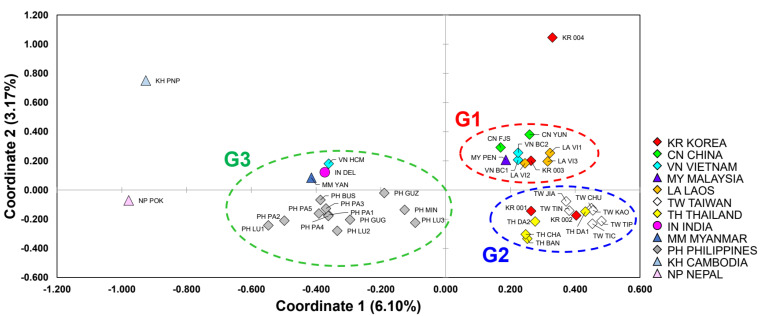
Principal coordinates analysis (PCoA) plotted by microsatellite data from 39 populations of *Bactrocera dorsalis* from 12 Asian countries by GENALEX. The *x*-axis is coordinate 1, ranging from −1.20 to 0.60, and the *y*-axis coordinate 2, from −0.60 to 1.20. CN SHA was excluded from the analysis because it was located outside due to the effect of null alleles.

**Figure 4 insects-12-00851-f004:**
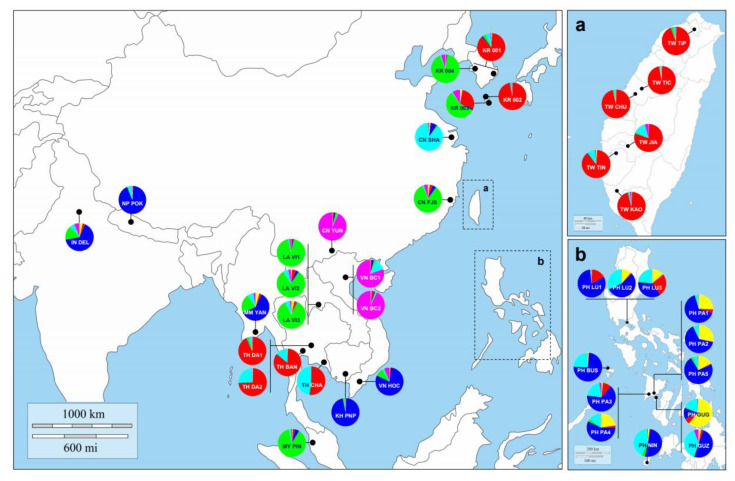
Bayesian clustering results (STRUCTURE for *K* = 6) in combination with the geographic locations of population collections. Each color means the independent assignment estimated by STRUCTURE. Dotted rectangles expand to (**a**) the Taiwan region and to (**b**) the Philippines region. KR, Korea; TW, Taiwan; CN, China; VN, Vietnam; LA, Laos; MM, Myanmar; TH, Thailand; KH, Cambodia; IN, India; NP, Nepal; PH, the Philippines; MY, Malaysia.

**Table 1 insects-12-00851-t001:** Summary of statistics for microsatellite loci from *B. dorsalis* populations. The number of individuals (No.); genotypic diversity (*GD*); observed heterozygosity (*H_o_*); expected heterozygosity (*H_e_*); Hardy–Weinberg equilibrium; gene diversity (*H_s_*); mean number of alleles (*N_a_*); allelic richness (*R_s_*); Weir and Cockerham’s estimates of inbreeding coefficient (*F_is_*); ns, non-significance in HWE; * *p*-values for heterozygote excess or deficit (*p* < 0.0001).

Population ID	No.	*GD*	*N_a_*	*N_e_*	*H_o_* (± SD)	*H_e_* (± SD)	HWE *	*H_s_*	*R_s_*	*F_is_*
KR 001	9	1	5.27	3.93	0.681 (0.313)	0.672 (0.216)	ns	0.71	2.73	0.05
KR 002	10	1	4.60	3.38	0.704 (0.240)	0.636 (0.161)	ns	0.67	2.55	−0.05
KR 003	5	1	5.07	4.16	0.667 (0.247)	0.691 (0.216)	ns	0.78	2.95	0.15
KR 004	2	1	1.67	1.62	0.500 (0.500)	0.283 (0.277)	ns	0.32	1.67	−0.58
TW CHU	7	1	4.40	3.27	0.571 (0.310)	0.589 (0.242)	ns	0.63	2.49	0.11
TW JIA	8	1	6.87	4.98	0.600 (0.223)	0.751 (0.130)	ns	0.82	3.06	0.26
TW KAO	20	1	8.73	4.93	0.657 (0.234)	0.695 (0.238)	ns	0.71	2.79	0.08
TW TIC	20	1	10.07	4.66	0.653 (0.177)	0.740 (0.144)	ns	0.76	2.90	0.14
TW TIN	20	1	10.20	5.30	0.573 (0.173)	0.731 (0.185)	ns	0.75	2.90	0.24
TW TIP	15	1	8.07	4.37	0.693 (0.201)	0.682 (0.194)	ns	0.71	2.73	0.02
CN FJS	24	1	7.27	4.27	0.522 (0.204)	0.677 (0.230)	ns	0.69	2.68	0.25
CN SHA	1	1	1.33	1.33	0.400 (0.507)	0.200 (0.254)	NA	NA	NA	NA
CN YUN	24	1	7.13	3.66	0.536 (0.231)	0.605 (0.238)	ns	0.62	2.47	0.14
VN BC1	18	1	8.40	4.63	0.400 (0.193)	0.678 (0.250)	ns	0.71	2.76	0.43
VN BC2	13	1	6.73	3.62	0.378 (0.254)	0.650 (0.176)	ns	0.69	2.63	0.45
VN HCM	18	1	6.53	3.34	0.581 (0.267)	0.649 (0.196)	ns	0.67	2.57	0.13
LA VI1	24	1	7.60	3.68	0.514 (0.212)	0.648 (0.225)	ns	0.67	2.60	0.23
LA VI2	24	1	8.20	3.55	0.489 (0.220)	0.632 (0.237)	ns	0.65	2.56	0.25
LA VI3	24	1	8.40	3.95	0.461 (0.193)	0.661 (0.221)	ns	0.68	2.64	0.32
MM YAN	25	1	8.67	4.44	0.518 (0.224)	0.723 (0.136)	ns	0.74	2.81	0.30
TH BAN	5	1	3.93	3.17	0.640 (0.275)	0.639 (0.132)	ns	0.72	2.65	0.11
TH CHA	4	1	3.87	3.27	0.700 (0.215)	0.665 (0.109)	ns	0.77	2.80	0.09
TH DA1	20	1	7.87	4.49	0.632 (0.232)	0.703 (0.194)	ns	0.72	2.78	0.13
TH DA2	23	1	6.87	4.07	0.624 (0.219)	0.715 (0.117)	ns	0.74	2.78	0.16
KH PNP	1	1	1.40	1.40	0.400 (0.507)	0.200 (0.254)	NA	NA	NA	NA
IN DEL	9	1	5.53	4.02	0.559 (0.228)	0.702 (0.142)	ns	0.76	2.81	0.26
NP POK	1	1	1.40	1.40	0.400 (0.507)	0.200 (0.254)	NA	NA	NA	NA
PH BUS	4	1	3.40	2.78	0.567 (0.312)	0.619 (0.100)	ns	0.74	2.62	0.23
PH GUG	24	1	8.20	3.35	0.559 (0.221)	0.641 (0.147)	ns	0.66	2.54	0.15
PH GUZ	8	1	5.60	3.79	0.536 (0.226)	0.685 (0.151)	ns	0.75	2.78	0.28
PH LU1	6	1	3.73	2.78	0.511 (0.299)	0.548 (0.249)	ns	0.61	2.38	0.16
PH LU2	13	1	6.87	4.48	0.557 (0.272)	0.708 (0.155)	ns	0.75	2.81	0.25
PH LU3	28	1	10.00	4.60	0.580 (0.169)	0.739 (0.136)	ns	0.76	2.87	0.23
PH MIN	13	1	6.40	4.13	0.582 (0.188)	0.686 (0.173)	ns	0.72	2.74	0.19
PH PA1	6	1	4.80	3.78	0.549 (0.302)	0.676 (0.148)	ns	0.76	2.79	0.28
PH PA2	22	1	7.13	3.83	0.524 (0.238)	0.677 (0.163)	ns	0.70	2.66	0.25
PH PA3	8	1	5.60	3.99	0.633 (0.234)	0.671 (0.178)	ns	0.72	2.74	0.12
PH PA4	9	1	5.80	3.71	0.566 (0.209)	0.691 (0.115)	ns	0.74	2.77	0.24
PH PA5	33	1	7.40	3.88	0.590 (0.190)	0.701 (0.105)	ns	0.71	2.68	0.17
MY PEN	24	1	7.07	3.41	0.481 (0.222)	0.619 (0.217)	ns	0.64	2.48	0.24

**Table 2 insects-12-00851-t002:** Analysis of molecular variance (AMOVA) of 565 individuals in 40 populations of *B. dorsalis* in Korea and Southeast Asia. Case 1: KR 001-004 vs. East Asian pop. Case 2, 3: Based on STRUCTURE (*K* = 2, *K* = 3) and PCoA results, ideal group difference estimation. Case 4: Inference of group variability according to sampling year. Case 5: Based on the longitude 113° E, west side (CH YUN, IN, KH, LA, MM, MY, NP, TH and VN) vs. east side (CH FJS, CH SHA, KR and PH).

Case	Source of Variation	d.f.	Sum of Squares	Variance Components	Percentage of Variation	*p*
1	Korea vs. East Asian country					
	Among groups	1	19.31	0.03550	0.61	<0.0001
	Among populations within groups	38	857.701	0.61627	10.61	<0.0001
	Within populations	1090	5620.326	5.15626	88.78	0.2176
	Total	1129	6497.337	5.80803		
2	Based on the STRUCTURE *K* = 2 scenario					
	Among groups	1	143.959	0.27097	4.57	<0.0001
	Among populations within groups	38	733.053	0.50821	8.56	<0.0001
	Within populations	1090	5620.326	5.15626	86.87	<0.0001
	Total	1129	6497.337	5.93544		
3	Based on the PCoA and *K* = 3 scenario in STRUCTURE analysis					
	Among groups	2	310.919	0.36108	6.13	<0.0001
	Among populations within groups	37	566.092	0.37043	6.29	<0.0001
	Within populations	1090	5620.326	5.15626	87.58	<0.0001
	Total	1129	6497.337	5.88777		
4	Based on sampling year, 2015 vs. 2016 vs. 2017					
	Among groups	2	188.855	0.19165	3.28	<0.0001
	Among populations within groups	37	688.157	0.49202	8.43	<0.0001
	Within populations	1090	5620.326	5.15626	88.29	<0.0001
	Total	1129	6497.337	5.83993		
5	Based on the longitude 113° E, west side vs. east side					
	Among groups	1	78.265	0.09072	1.56	<0.0001
	Among populations within groups	38	798.746	0.57314	9.85	<0.0001
	Within populations	1090	5620.326	5.15626	88.59	0.0036
	Total	1129	6497.337	5.82012		

**Table 3 insects-12-00851-t003:** The most likely demographic scenario for *B. dorsalis* by the DIYABC analysis ‘A’ (see Appendix A). Scenario 1: Subgroup 1 (CN + VN) as an ancestor group, Subgroup 2 (TW + TH) first separated from Subgroup 1 (CN + VN), then Subgroup 3 (PH + MY + IN) diverged from Subgroup 2 (TW + TH) (Appendix A). Scenario 2: ancestor group of Subgroup 1 (CH + VN), similar to Scenario 1, but reversed the diverging order of Subgroup 2 (TW + TH) and Subgroup 3 (PH + MY + IN) (Appendix A). Scenarios 3 and 4: Subgroup 2 (TW + TH) as the ancestor group, and the diverging order of Subgroup 1 (CH + VN) and Subgroup 3 (PH + MY + IN) (Appendix A). Scenarios 5 and 6: Subgroup 3 (PH + MY + IN) as the ancestor group, and the diverging order of Subgroup 1 (CH + VN) and Subgroup 2 (TW + TH) (Appendix A). Values in bold indicate the first and second best scenarios in the analysis ‘A’.

Logistic Regression Closest	Scenario 1	Scenario 2	Scenario 3	Scenario 4	Scenario 5	Scenario 6
6000	0.0495 [0.0302, 0.0689]	0.0558 [0.0351, 0.0765]	0.2944 [0.2495, 0.3393]	0.2907 [0.2464, 0.3350]	0.1674 [0.1331, 0.2017]	0.1422 [0.1094, 0.1750]
12,000	0.0448 [0.0317, 0.0579]	0.0495 [0.0361, 0.0628]	0.2954 [0.2623, 0.3286]	0.2976 [0.2644, 0.3307]	0.1673 [0.1421, 0.1925]	0.1454 [0.1211, 0.1698]
18,000	0.0426 [0.0322, 0.0529]	0.0468 [0.0363, 0.0573]	0.2982 [0.2703, 0.3261]	0.3031 [0.2750, 0.3312]	0.1644 [0.1436, 0.1852]	0.1450 [0.1248, 0.1652]
24,000	0.0414 [0.0325, 0.0503]	0.0469 [0.0377, 0.0561]	0.2994 [0.2747, 0.3241]	0.3059 [0.2809, 0.3308]	0.1621 [0.1440, 0.1802]	0.1443 [0.1266, 0.1620]
30,000	0.0398 [0.0321, 0.0476]	0.0472 [0.0389, 0.0555]	0.3003 [0.2779, 0.3227]	0.3076 [0.2848, 0.3304]	0.1611 [0.1448, 0.1774]	0.1439 [0.1280, 0.1599]
36,000	0.0388 [0.0319, 0.0458]	0.0473 [0.0397, 0.0549]	0.3011 [0.2803, 0.3219]	0.3093 [0.2882, 0.3305]	0.1597 [0.1448, 0.1747]	0.1437 [0.1290, 0.1584]
42,000	0.0378 [0.0315, 0.0442]	0.0474 [0.0404, 0.0545]	0.3014 [0.2820, 0.3208]	0.3111 [0.2913, 0.3309]	0.1588 [0.1448, 0.1727]	0.1434 [0.1297, 0.1572]
48,000	0.0370 [0.0312, 0.0429]	0.0477 [0.0411, 0.0543]	0.3019 [0.2835, 0.3202]	0.3119 [0.2931, 0.3306]	0.1582 [0.1451, 0.1713]	0.1433 [0.1304, 0.1563]
54,000	0.0366 [0.0311, 0.0420]	0.0477 [0.0414, 0.0539]	0.3014 [0.2840, 0.3188]	0.3125 [0.2947, 0.3303]	0.1583 [0.1458, 0.1707]	0.1436 [0.1313, 0.1559]
60,000	0.0361 [0.0310, 0.0412]	0.0476 [0.0416, 0.0535]	**0.3008 [0.2842, 0.3174]**	**0.3135 [0.2965, 0.3306]**	0.1581 [0.1463, 0.1700]	0.1439 [0.1321, 0.1556]

**Table 4 insects-12-00851-t004:** The most likely demographic scenario by the DIYABC analysis ‘B’ to trace the origin of the *B. dorsalis* quarantine-detected groups on the Korean Peninsula (see Appendix A). Subgroup 4 to KR 001, 002 or 003, and each analysis was performed to trace the origin of each Korean group’s introduction route. Subgroup 4 set to KR 001 in Analysis B-1, KR 002 in B-2 and KR 003 in B-3. Scenario 1: Subgroup 4 introduced from Subgroup 1 (CN + VN). Scenario 2: Subgroup 4 introduced from Subgroup 3 (PH + MY + IN). Scenario 3: Subgroup 4 introduced from Subgroup 2 (TW + TH). In Analysis B-1, Scenario 3 was most suitable for KR 001 (Appendix A). Values in bold indicate the best scenario in each analysis.

Analysis	Subgroup 1	Subgroup 2	Subgroup 3	Subgroup 4	Scenario 1	Scenario 2	Scenario 3
B-1	CN + VN	TW + TH	PH + MY + IN	KR 001	0.0050 [0.0026, 0.0074]	0.0015 [0.0007, 0.0023]	**0.9935 [0.9907, 0.9964]**
B-2	CN + VN	TW + TH	PH + MY + IN	KR 002	0.0040 [0.0026, 0.0053]	0.0023 [0.0014, 0.0032]	**0.9937 [0.9919, 0.9956]**
B-3	CN + VN	TW + TH	PH + MY + IN	KR 003	**0.9552 [0.9458, 0.9646]**	0.0428 [0.0335, 0.0521]	0.0020 [0.0011, 0.0029]

## Data Availability

Data are available upon request from the authors.

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
