# Peer review of "Population Genetics for Inferring Introduction Sources of the Oriental Fruit Fly, Bactrocera dorsalis: A Test for Quarantine Use in Korea"

_insects, 2021, doi:10.3390/insects12100851_

Round 1

Reviewer 1 Report

The manuscript is well written. The research question is interesting with some relevant background and literature provided. The methods are quite clear and include most of the required information. 

Here are some comments.

Line 30. ABC should be defined in full. 

Lines 119-121. The authors refer to have studied individuals from 36 geographically different collections and four quarantine collections. In Table S1 they state 36 Pop. ID (one per line) but some populations have the same name of collection site and Latitude, longitude, differing only by date of collection. Thus, they are not geographically different collections.

Line 126. Why is there no data about the samples assigned to the quarantine-detected groups 1, 2, 3 and 4 collected from South Korea? When were they collected? Were in years close to those of the other samples (2015-2017)?

Line 140. Is this reference correct? In the cited article, molecular identification was not performed.

Figure 1. It would be good to clarify the collection locations so that each point corresponds to a location described in Table S1. Maybe collection points on Taiwan Island and Philippine Islands should only be shown on enlarged maps.

Lines 155-156. The primers described by Folmer primers (reference 64) are different from these primers.

Line 293. It would be good to start with the total number of haplotypes obtained (50?).

Line 304. What about H14 and H38? 

Line 590. Why did the authors consider KR004 as having an identical population and genetic structure in Fujian, China or Laos? In Figure 3, KR004 seems to me very distant from the G1 group.

Line 614. FTA should be defined in full

It would be interesting if the authors had discussed the relationship between the use of samples from a species complex and the variability of the markers, for example, is it possible that the variability found is related to subspecies?

Does the quantity of samples from the 4 quarantine groups are good representants of the invasive samples found in the period and in that region?

Author Response

REVIEWER1 – comments and responses

The manuscript is well written. The research question is interesting with some relevant background and literature provided. The methods are quite clear and include most of the required information. 

>[response] Thank you very much for your invaluable comments. We have corrected the comments you pointed out as much as possible.

Here are some comments.

Line 30. ABC should be defined in full. 

>[response] corrected to “approximate Bayesian computation” in line 30

Lines 119-121. The authors refer to have studied individuals from 36 geographically different collections and four quarantine collections. In Table S1 they state 36 Pop. ID (one per line) but some populations have the same name of collection site and Latitude, longitude, differing only by date of collection. Thus, they are not geographically different collections.

>[response] corrected as “36 geographically different collections -> 36 geographically or temporally different (if in the same location) collections” in line 127

Line 126. Why is there no data about the samples assigned to the quarantine-detected groups 1, 2, 3 and 4 collected from South Korea? When were they collected? Were in years close to those of the other samples (2015-2017)?

>[response] Sampling data about the samples of the quarantine-detected groups 1, 2, 3 and 4 collected from South Korea were added as collection site, number of samples and date in Table S1. However, exact GPS coordinates of the location is not permitted because exposure to quarantine activities is impossible for security. Even open collection data is considered to be sufficient evidence for scientific use.

Line 140. Is this reference correct? In the cited article, molecular identification was not performed.

>[response] It was an mistake. The molecular identification reference was changed to

“71(formely 64). Hebert, P.D.N.; Penton, E.H.; Burns, J.M.; Janzen, D.H.; Hallwachs, W. Ten species in one: DNA barcoding reveals cryptic species in the neotropical skipper butterfly Astraptes fulgerator. Proceedings of the National Academy of Sciences of the United States of America 2004, 101, 14812-14817, doi:DOI 10.1073/pnas.0406166101.” in line 150

Figure 1. It would be good to clarify the collection locations so that each point corresponds to a location described in Table S1. Maybe collection points on Taiwan Island and Philippine Islands should only be shown on enlarged maps.

>[response] Figure 1 is modified as indicating number of collections at each country. The sentence is added in legend of Figure1, “Number of collections at each country were indicated in parenthesis.” Because 36 collections were geographically or temporally different (if in the same location), some of them could not be pointed on map (same point). Detailed location information is described in Table S1. In line 159

Lines 155-156. The primers described by Folmer primers (reference 64) are different from these primers.

>[response] It was an mistake. That reference was changed to

“71(formely 64). Hebert, P.D.N.; Penton, E.H.; Burns, J.M.; Janzen, D.H.; Hallwachs, W. Ten species in one: DNA barcoding reveals cryptic species in the neotropical skipper butterfly Astraptes fulgerator. Proceedings of the National Academy of Sciences of the United States of America 2004, 101, 14812-14817, doi:DOI 10.1073/pnas.0406166101.” in line 166-168.

Line 293. It would be good to start with the total number of haplotypes obtained (50?).

>[response] In line 305-306, It was changed to “Built a crown-like MJ network, totally 50 haplotypes (H1-H50) were obtained (Figure 2). There were some high-frequency haplotypes (such as H1, H2, H3, H23, H32 and H31) located in the center, and other rare haplotypes were connected to them through several mutation steps.”

Line 304. What about H14 and H38? 

>[response] In line 314, “H14 (CN) and H38 (PH) were further isolated with 8 steps from H1.”

Line 590. Why did the authors consider KR004 as having an identical population and genetic structure in Fujian, China or Laos? In Figure 3, KR004 seems to me very distant from the G1 group.

>[response] In line 601-603, “On the other hand, KR 003 had identical populations and genetic structure in Fujian, China or Laos (Figures 3 and 4). Although K004 was similarly clustered to in Fujian, China or Laos (Figure 4), it could not be estimated from where it originated due to large difference genetically from other groups (Figure 3).”

Line 614. FTA should be defined in full

>[response] corrected to “Free Trade Agreement” in 626

It would be interesting if the authors had discussed the relationship between the use of samples from a species complex and the variability of the markers, for example, is it possible that the variability found is related to subspecies?

>[rebuttal] Unfortunately, it is impossible to confirm the variability based on member of species complex (i.e. subspecies as papayae, philippinensis, invadens, etc.). There is no information on host or other subspecific characteristics when we captured those samples. To capture B. dorsalis, we equipped a transparent cylindrical acrylic trap with methyl eugenol attractants and insecticide-infused cotton. However, we made “Allelic frequencies based on the polymorphisms of 15 microsatellite markers at each study site.” in Table S7, which can be useful to confirm the regional variabilities between our collections.

Does the quantity of samples from the 4 quarantine groups are good representants of the invasive samples found in the period and in that region?

>[response] Sampling data about the samples of the quarantine-detected groups 1, 2, 3 and 4 collected from South Korea were added as collection site, number of samples and date in Table S1. However, exact GPS coordinates of the location is not permitted because exposure to quarantine activities is impossible for security. Even open collection data is considered to be sufficient evidence for scientific use.

Reviewer 2 Report

Dr. Kim and Collaborators obtained barcode sequences and microsatellite genotypes from roughly 500 individuals of B. dorsalis from South East Asia. They use different analytical strategies (F statistics, bayesian clustering, assignment of genotypes, ABC) to investigate the genetic structure of populations in the area with a specific attention to the possibility to identify the source area of invasive Korean individuals.

I think that the authors have a neat dataset, with a sufficient number of samples from a large and biogeographically meaningful area. This puts them in the correct position to address these issues effectively. The subject matter of the study is of significant importance, not only for our understanding of the evolution of one species, but also for the implications that knowing the genetic structure of B. dorsalis and being able to assign unknown samples to source populations may have in terms of plant protection and reduction of economic losses. This is a significant added value of this study.

Nevertheless, as a consequence of this latter point, there has been a lot of attention by the scientific community and decision makers alike on B. dorsalis. This calls for an extra effort towards clarity and a correct interpretation/presentation of the results.

MAJOR:

The authors clearly acknowledge that there is (or there has been) some confusion in the identification of the natural assemblage generally regarded as B. dorsalis. The authors discuss this at lines 47-56 and 132-136, in an attempt to be very clear about the actual precinct of their research. Nevertheless, the variable meaning that species names took through time and in the words of different authors, exacerbated by a language that is not always easy to understand, makes reading of these parts anything but clear. Sentences 'B.dorsalis s. str. sensu Boykin' (line 55), 'Bactrocera dorsalis complex s. str [12,13]' (line 135) and 'belonging to the speices complex' (line 136) are difficult to reconcile with the clear cut observation that, based on the current literature, B. dorsalis is a bona fide species. I strongly suggest a paragraph where taxonomic entities, as they have been intended by differnt authors, are clarified, and a clear statement is given about what this study is about.
This opens to an ancillary issue at lines 137-142. It is stated that collected samples were analysed for morphology and haplotype sequences before genotyping. I do not quite understand the meaning of haplotype sequencing in this context. Were haplotype sequences used to complement morphology in the sorting of a complex sample (possibly including other species or other dorsalis-related entities)? Did this led to the exclusion of some samples? 

The nature of samples KR001-4, defined as 'quarantine collections' at Korean borders, is unclear and may open to some issues. Throughout the analysis, thay are treated just like all other populations, but they may not be 'populations' in the biological sense. The Authors should clarify their nature and modify the analysis to address this issue. As a general idea, if the 4 samples (of 2-10 individuals) represent unique outbreaks, implying that the 2-10 individuals from a sample are expected to be genetically related (and related to a population of origin), they may indeed be treated as a populations, ableit with serious concerns about their being at equilibrium, that is a requisite for most of the methods used. If, on the other hand, they are non natural assemblages (i.e. individuals collected as the same port of entry, but not knowing if they come from one and the same population of origin), they should not be treated as populations during the analysis. One possible solution, at least for the structure analysis, would be to conduct the analysis without 'quarantine collections' and then conduct an assignment test on individual (quarantined) flies asking which is the likely source population. This would relax the need for the flies of each of these 4 groups to be natural assemblages, as the 'object' being assigned would be the individual and not the population.  

The language is not always clear, and a number of passages are difficult to interpret with the due precision. I strongly advise the manuscript is revised by a native English speaker.

MINOR:
Figure 1. The number of flags per state does not always match the expected number of samples form the state (e.g. 1 in Vietnam, Laos and Thailand, where 3, 3, 4 are expected). Please redraw.

Lines 211-223. Which 'populations' are used for these analyses? Are samples from each and every location treated as a 'population'?

Troughout the manuscript, reference is made to multiple STRUCTURE clustering levels (2, 3, 6). While it is fully possible that different k expose different features in the data, and therefore clustering under multiple k can be shown and discussed, the Authors have conducted a standard analysis (Evanno method) to identify the correct k and are expected to show the clustering under that k. If alternative k are used, this should be clear to the reader and a justification should be given. 

A number of different hypotheses on the origin and population structure of the species have been made by different authors in the past. These are nevertheless not introduced to the reader until the Discussion section. While I think the Discussion section is the appropriate place for a detailed comparison of the research outcomes with previous accounts (as the Authors do), I suggest that the main hypotheses and main open questions should be introduced in the Introduction section.

Line 93: 'induce', please reword.
Line 155: reword.
Line 182, 187: 'availability' please restate.
Line 186: 'fitness test' pease clarify.
Lines 195-196, 201-203: please clarify.
Line 234: STRUCTURE is about clustering individuals, not populations, please reword.
Line 294: 465 individuals were barcoded out of 565. Which is the reason for the difference? 

The manuscript is rather long and some sentences are overabundant. Considering that it will need to be revised for correct English syntax, I would encourage the Authors to shorten the text altogether. I think that the manuscript could be shortened by 20% with no detriment to completeness and clarity. 

Author Response

REVIEWER2 – comments and responses

Dr. Kim and Collaborators obtained barcode sequences and microsatellite genotypes from roughly 500 individuals of B. dorsalis from South East Asia. They use different analytical strategies (F statistics, bayesian clustering, assignment of genotypes, ABC) to investigate the genetic structure of populations in the area with a specific attention to the possibility to identify the source area of invasive Korean individuals.

I think that the authors have a neat dataset, with a sufficient number of samples from a large and biogeographically meaningful area. This puts them in the correct position to address these issues effectively. The subject matter of the study is of significant importance, not only for our understanding of the evolution of one species, but also for the implications that knowing the genetic structure of B. dorsalis and being able to assign unknown samples to source populations may have in terms of plant protection and reduction of economic losses. This is a significant added value of this study.

Nevertheless, as a consequence of this latter point, there has been a lot of attention by the scientific community and decision makers alike on B. dorsalis. This calls for an extra effort towards clarity and a correct interpretation/presentation of the results.

>[response] Thank you very much for your invaluable comments. We have corrected the comments you pointed out as much as possible.

MAJOR:

The authors clearly acknowledge that there is (or there has been) some confusion in the identification of the natural assemblage generally regarded as B. dorsalis. The authors discuss this at lines 47-56 and 132-136, in an attempt to be very clear about the actual precinct of their research. Nevertheless, the variable meaning that species names took through time and in the words of different authors, exacerbated by a language that is not always easy to understand, makes reading of these parts anything but clear. Sentences 'B.dorsalis s. str. sensu Boykin' (line 55), 'Bactrocera dorsalis complex s. str [12,13]' (line 135) and 'belonging to the speices complex' (line 136) are difficult to reconcile with the clear cut observation that, based on the current literature, B. dorsalis is a bona fide species. I strongly suggest a paragraph where taxonomic entities, as they have been intended by differnt authors, are clarified, and a clear statement is given about what this study is about.
This opens to an ancillary issue at lines 137-142. It is stated that collected samples were analysed for morphology and haplotype sequences before genotyping. I do not quite understand the meaning of haplotype sequencing in this context. Were haplotype sequences used to complement morphology in the sorting of a complex sample (possibly including other species or other dorsalis-related entities)? Did this led to the exclusion of some samples? 

>[response] Historically, B. dorsalis was a species with many taxonomic issues. Their sister species have biologically different host plants, but they are very similar morphologically, causing a lot of confusion. In addition, a recent study revealed that regionally different biotypes are one species from a phylogenetic point of view. Of course, as you said, B. dorsalis is a bona fide species. However, if there is no mention of the taxonomic history of B. dorsalis when it is understood by researchers dealing with B. dorsalis, it is true that it is unclear at what boundary the species is biologically located with only the nomenclature of B. dorsalis itself. To clarify this taxonomic problem, we explain the classification of B. dorsalis in the introduction, and finally show the exact identification of the valid species. Since the taxonomic problem has been clarified, there are no problems with morphology and molecular identification in this study. Based on the review, it has been modified as follows.

Line 57-58: In this study, we focused on B. dorsalis redefined by the previous studies [12,13] for tracing the origin of the invasive source.

Line 143-144: Accordingly, ‘Bactrocera dorsalis’, which is mentioned after that, is meant to the valid species taxonomically revised by the two recent studies [12,13].

The nature of samples KR001-4, defined as 'quarantine collections' at Korean borders, is unclear and may open to some issues. Throughout the analysis, thay are treated just like all other populations, but they may not be 'populations' in the biological sense. The Authors should clarify their nature and modify the analysis to address this issue. As a general idea, if the 4 samples (of 2-10 individuals) represent unique outbreaks, implying that the 2-10 individuals from a sample are expected to be genetically related (and related to a population of origin), they may indeed be treated as a populations, ableit with serious concerns about their being at equilibrium, that is a requisite for most of the methods used. If, on the other hand, they are non natural assemblages (i.e. individuals collected as the same port of entry, but not knowing if they come from one and the same population of origin), they should not be treated as populations during the analysis. One possible solution, at least for the structure analysis, would be to conduct the analysis without 'quarantine collections' and then conduct an assignment test on individual (quarantined) flies asking which is the likely source population. This would relax the need for the flies of each of these 4 groups to be natural assemblages, as the 'object' being assigned would be the individual and not the population.

>[response & rebuttal] First, sampling data about the samples of the quarantine-detected groups 1, 2, 3 and 4 collected from South Korea were added as collection site, number of samples and date in Table S1. Second, because the topic of this study is inferring the introduction sources of the oriental fruit fly, Bactrocera dorsalis, for quarantine use in Korea, it is impossible to exclude the quarantine-detected group from analysis such as STRUCTURE, and it is an analysis that must be performed to compare it with samples from other regions. Although all these analysis techniques of population genetics are designed to analyze 'natural populations', many previous studies have shown that they are also useful for analyzing ‘pooled non-natural group’ samples by further extending the concept of 'natural populations'. Although it seems that goes against the principle of population genetics, this study is meaningful in inferring the country of origin using the technique.

With regard to excluding K001-004 from STRUCTURE analysis, STRUCTURE analysis of all samples is necessary to estimate genetically all samples for identifying their genetic assignment of all individuals in populations. This technique is used to identify genetic assignment of individual as well as that of population. In addition, to perform a DIYABC, we need to make an appropriate scenario based on the results of STRUCTURE analysis. Therefore, it is impossible to exclude those samples as this is an important flow in the analysis process.

To avoid confusion about 'pooled non-natural group' and 'natural populations', I put the following sentence:

Line 135-138: Although these quarantine-detected groups as pooled non-natural ones were not like other naturally collected populations, we set them as four ‘virtual’ populations to infer their origins and invasion routes compared with foreign samples.

The language is not always clear, and a number of passages are difficult to interpret with the due precision. I strongly advise the manuscript is revised by a native English speaker.

>[response] Since the authors are not English native, I admit that there is a little awkward expression in the description, but it has been confirmed that there is no big problem in delivering the results. In addition, this manuscript was proofread and edited by the professional native English editors of Scientific English Research Paper Editing Service at ENAGO (www.enago.co.kr). We have attached the certificate of editing as supporting information (see last page).

 MINOR:
Figure 1. The number of flags per state does not always match the expected number of samples form the state (e.g. 1 in Vietnam, Laos and Thailand, where 3, 3, 4 are expected). Please redraw.

>[response] Figure 1 is modified as indicating number of collections at each country. The sentence is added in legend of Figure1, “Number of collections at each country were indicated in parenthesis.” Because 36 collections were geographically or temporally different (if in the same location), some of them could not be pointed on map (same point). Detailed location information is described in Table S1.

Lines 211-223. Which 'populations' are used for these analyses? Are samples from each and every location treated as a 'population'?

>[response] Populations used in these analyses are 36 geographically or temporally different (if in the same location) collections and four quarantine-detected groups. I was already addressed why quarantine-detected groups treated as pooled non-natural ones in this study. Based on the review, it has been modified as follows.

Line 223: Using 565 B. dorsalis individual samples from 40 collections (Figure 1; Table S1), we performed the data processing for calculating allele frequency …

Throughout the manuscript, reference is made to multiple STRUCTURE clustering levels (2, 3, 6). While it is fully possible that different k expose different features in the data, and therefore clustering under multiple k can be shown and discussed, the Authors have conducted a standard analysis (Evanno method) to identify the correct k and are expected to show the clustering under that k. If alternative k are used, this should be clear to the reader and a justification should be given. 

>[response] I agree your review. Why alternative K was used is explained as follows.

Line 380-383: Although K = 2 was the best, the results of K = 3 and K = 6 were referred thereafter because those were effective to display the relationships between the populations when compared and combined with the other analyses such as PCoA.

A number of different hypotheses on the origin and population structure of the species have been made by different authors in the past. These are nevertheless not introduced to the reader until the Discussion section. While I think the Discussion section is the appropriate place for a detailed comparison of the research outcomes with previous accounts (as the Authors do), I suggest that the main hypotheses and main open questions should be introduced in the Introduction section.

>[response] I agree your review. We insert the arguments about the ancestry origin of B. dorsalis in introduction as follows:

Line 73-78: “With regard to the ancestry origin of B. dorsalis, some previous studies suggest that it was first originated from the mainland China or Taiwan [30-34] and may have migrated to the west and spread to southeast Asia [30,35,36], while the other studies showed that India, Nepal and its neighboring countries are the most ancestry origin of B. dorsalis spreading to the south and east Asia [10,37]. These conflicting theories regarding the ancestry origin of B. dorsalis have not yet been clearly concluded.”

Line 93: 'induce', please reword.

>[response] rewording the sentence as “Microsatellite markers generally have short tandem repeats with fast mutation rates …” in line 100.

Line 155: reword.

>[response] rewording the sentence as The primer set of LepF1 (5’-ATTCAACCAATCATAAAGATATTGG-3’) and LepR1 (5’-TAAACTTCTGGATGTCCAAAAAATCA-3’) was used to amplify the COI barcode sequence [71]. In Line 166-168

Line 182, 187: 'availability' please restate.

>[response] rewording the expression instead of 'availability' as.

“… investigate the amplification success and polymorphism …” line 194

“… estimated the amplification success and polymorphism …” line 199

Line 186: 'fitness test' please clarify.
>[response] rewording as: “We conducted a total of 172 PCR tests …” line 198

Lines 195-196, 201-203: please clarify.

>[response] in line 208a negative dye of “ deleted

In line 232, “After visualization of fluorescent-labeled we carried out PCR products in the same manner as mentioned above.” deleted. It is a redundancy of “using fluorescent-labeled (5’ modification) forward primers (in line 214, 215).

Line 234: STRUCTURE is about clustering individuals, not populations, please reword.

>[response] In Line 275: “We used STRUCTURE 2.3.4 to analyze the genetic structure of 565 B. dorsalis individuals from 40 populations using a Bayesian clustering approach [89].”

Line 294: 465 individuals were barcoded out of 565. Which is the reason for the difference? 

>[response] We did not perform COI barcode sequencing of all samples from all collections because it was expected that a lot of duplicate sequences such as H1 would be generated in samples from some regions. It is also included the issue of experimental budgets. However, there is no problem that the sample size of microsatellite and COI barcode is different to interpret the results.

The manuscript is rather long and some sentences are overabundant. Considering that it will need to be revised for correct English syntax, I would encourage the Authors to shorten the text altogether. I think that the manuscript could be shortened by 20% with no detriment to completeness and clarity. 

 >[response] Considering the sample size and analysis methodology of this study, the volume of manuscript is considered appropriate. Except for materials & methods and references, it doesn't take up a lot of pages. It has the average volume of population genetics papers. We would like to inform you that the content was written in compliance with the submission guidelines of the 'Insects' journal site.

Reviewer 3 Report

This is a very comprehensive study of Bactrocera dorsalis over a wide geographic area and investigates the origin of this invasive species to Korea using microsatellite markers and mitochondrial DNA. Analyses and overall conclusions appear sound. Some recommendations are made to improve organisation and clarity of the paper.

  1.  Methods: Line 123 onwards. Collection locations are called populations (pops.), but in a genetic paper such as this, they should really be described as geographic collection locations before data on genetic population structure have been obtained.
  2. Line 136: What implications does using a species complex have on the results and analysis. Were any of the samples screened out at this stage (line 140)?
  3. Line 186: Are the fitness tests, goodness of fit tests? What is their nature?
  4. Is there capacity to add further subheadings to the analysis section and strictly group analyses that are addressing each specific question?
  5. Line 323 - please define non-clonal MLGs
  6. Some of the Table headings are very long. It may be better to have some of the information as a footnote.
  7. Line 552 - the word 'migrated' has been used here, but I think 'were carried' might be more accurate in this case.
  8. Overall, if the results section could be made more succinct, this would help the reader.

Author Response

REVIEWER3 – comments and responses

This is a very comprehensive study of Bactrocera dorsalis over a wide geographic area and investigates the origin of this invasive species to Korea using microsatellite markers and mitochondrial DNA. Analyses and overall conclusions appear sound. Some recommendations are made to improve organisation and clarity of the paper.

>[response] Thank you very much for your invaluable comments. We have corrected the comments you pointed out as much as possible.

  1.  Methods: Line 123 onwards. Collection locations are called populations (pops.), but in a genetic paper such as this, they should really be described as geographic collection locations before data on genetic population structure have been obtained.

>[response] I agree your review. Reword populations (pops.) -> collections. There are some collections (one or two individuals) in which it is difficult to have a concept of population, we substituted a collection in this part.

  1. Line 136: What implications does using a species complex have on the results and analysis. Were any of the samples screened out at this stage (line 140)?

>[response] as same response to reviewer 1

Line 143-144: Accordingly, ‘Bactrocera dorsalis’, which is mentioned after that, is meant to the valid species taxonomically revised by the two recent studies [12,13].

We sampled but B. dorsalis, not B. dorsalis complex. It was a mistake to write its taxonomic status. Therefore ‘complex’ was deleted in the whole text. (line 150, 152)

  1. Line 186: Are the fitness tests, goodness of fit tests? What is their nature?

>[response] as same response to reviewer 2

rewording as: “We conducted a total of 172 PCR tests …” line 198

  1. Is there capacity to add further subheadings to the analysis section and strictly group analyses that are addressing each specific question?

>[response & rebuttal] Addressing each specific question of main objectives is addressed in order through the results and discussion content. Strict description of subheadings separately was not made in consideration of the increase in the length of the manuscript. In addition, reviewer 2 points out that the manuscript is too long. Please understand this.

  1. Line 323 - please define non-clonal MLGs

>[response] “multilocus genotypes (MLGs)” is already defined in line 227

revised as “non-clonal MLGs, i.e. non-identical genotypes estimated by multiple loci, (Table 1).” Line 334

  1. Some of the Table headings are very long. It may be better to have some of the information as a footnote.

>[response] According to your comment, redundancy and unnecessary descriptions on the legend of tables 3 and 4 have been greatly deleted and modified. Thank you.

  1. Line 552 - the word 'migrated' has been used here, but I think 'were carried' might be more accurate in this case.

>[response] corrected it as “were transferred” (line 541). Thank you.

  1. Overall, if the results section could be made more succinct, this would help the reader.

>[response] After several reviews, we tried to write the results as concisely as possible, and that's reflected in the results. Many parts have been revised and improved through the review process. Thank you for your understanding on this.

Round 2

Reviewer 2 Report

Dr Kim and Collaborators are resubmitting an emended version of their original mauscript. This latter version shows some improvements over the former, but I think that issues marked as MAJOR should have been dealt more thoroughly.

Specifically, I did not suggest that quarantine samples are removed from the analysis - something that the Authors find non acceptable given that this is the primary aim of their study - but, in turn, to conduct an alternative analysis that, in my view, may circumvent the problem: 'One possible solution, at least for the structure analysis, would be to conduct the analysis without 'quarantine collections' and then conduct an assignment test on individual (quarantined) flies asking which is the likely source population. This would relax the need for the flies of each of these 4 groups to be natural assemblages, as the 'object' being assigned would be the individual and not the population'.

I still think assignment tests on individual quarantine genotypes should be performed. 

The Editor, based on my as well as other reviews, is in the appropriate position to provide a final decision on this point.

Author Response

Reviewer 2 comment:

Dr Kim and Collaborators are resubmitting an emended version of their original mauscript. This latter version shows some improvements over the former, but I think that issues marked as MAJOR should have been dealt more thoroughly.

Specifically, I did not suggest that quarantine samples are removed from the analysis - something that the Authors find non acceptable given that this is the primary aim of their study - but, in turn, to conduct an alternative analysis that, in my view, may circumvent the problem: 'One possible solution, at least for the structure analysis, would be to conduct the analysis without 'quarantine collections' and then conduct an assignment test on individual (quarantined) flies asking which is the likely source population. This would relax the need for the flies of each of these 4 groups to be natural assemblages, as the 'object' being assigned would be the individual and not the population'.

I still think assignment tests on individual quarantine genotypes should be performed. 

The Editor, based on my as well as other reviews, is in the appropriate position to provide a final decision on this point.

Response:

Thank you for your review. Actually, I initially did assignment tests for individual quarantine samples collected from Korea but was not attached in the text (as preliminary result). As suggested by reviewer, I inserted the content of STURUCTER result of the 26 quarantine-detected individuals from Korea as follows:

Line 135-140.

The individuals of 26 Korean quarantine-detected samples were assigned to four groups (KR 001-004) by STRUCTURE analysis (K = 4) according to Evanno et al. [90] in Figure S1. Although these quarantine-detected groups as pooled non-natural ones were not like other naturally collected populations, we set them as four ‘virtual’ populations to infer their origins and invasion routes compared with foreign samples according to the STRUCTURE analysis (Figure S1).

Figure S1. The individual assignment result of 26 Korean samples from four quarantine-detected groups (KR 001-004) by STRUCTURE analysis (K = 4). According to Evanno et al. [90], the best delta K was estimated to 4. The ordering number of individuals are consistent to that in Table S1.
